# NICU Admission for Term Neonates in a Large Single-Center Population: A Comprehensive Assessment of Risk Factors Using a Tandem Analysis Approach

**DOI:** 10.3390/jcm11154258

**Published:** 2022-07-22

**Authors:** Shahar Talisman, Joshua Guedalia, Rivka Farkash, Tehila Avitan, Naama Srebnik, Yair Kasirer, Michael S. Schimmel, Dunia Ghanem, Ron Unger, Sorina Grisaru Granovsky

**Affiliations:** 1Shaare Zedek Medical Center, Department of Obstetrics & Gynecology, School of Medicine, Hebrew University, Jerusalem 9103102, Israel; shahar.talisman@gmail.com (S.T.); rivka_f@szmc.org.il (R.F.); srebnikn@szmc.org.il (N.S.); duniaghanem@gmail.com (D.G.); sorina@szmc.org.il (S.G.G.); 2The Mina and Everard Goodman Faculty of Life Sciences, Bar Ilan University, Ramat-Gan 5290002, Israel; joshuaguedalia@gmail.com (J.G.); ron@biocom1.ls.biu.ac.il (R.U.); 3Shaare Zedek Medical Center, Department of Pediatrics, School of Medicine, Hebrew University, Jerusalem 9103102, Israel; yairkasir@szmc.org.il (Y.K.); schimmel@szmc.org.il (M.S.S.)

**Keywords:** neonatal intensive care, term NICU, machine learning

## Abstract

Objective: Neonatal intensive care unit (NICU) admission among term neonates is associated with significant morbidity and mortality, as well as high healthcare costs. A comprehensive NICU admission risk assessment using an integrated statistical approach for this rare admission event may be used to build a risk calculation algorithm for this group of neonates prior to delivery. Methods: A single-center case–control retrospective study was conducted between August 2005 and December 2019, including in-hospital singleton live born neonates, born at ≥37 weeks’ gestation. Analyses included univariate and multivariable models combined with the machine learning gradient-boosting model (GBM). The primary aim of the study was to identify and quantify risk factors and causes of NICU admission of term neonates. Results: During the study period, 206,509 births were registered at the Shaare Zedek Medical Center. After applying the study exclusion criteria, 192,527 term neonates were included in the study; 5292 (2.75%) were admitted to the NICU. The NICU admission risk was significantly higher (ORs [95%CIs]) for offspring of nulliparous women (1.19 [1.07, 1.33]), those with diabetes mellitus or hypertensive complications of pregnancy (2.52 [2.09, 3.03] and 1.28 [1.02, 1.60] respectively), and for those born during the 37th week of gestation (2.99 [2.63, 3.41]; *p* < 0.001 for all), adjusted for congenital malformations and genetic syndromes. A GBM to predict NICU admission applied to data prior to delivery showed an area under the receiver operating characteristic curve of 0.750 (95%CI 0.743–0.757) and classified 27% as high risk and 73% as low risk. This risk stratification was significantly associated with adverse maternal and neonatal outcomes. Conclusion: The present study identified NICU admission risk factors for term neonates; along with the machine learning ranking of the risk factors, the highly predictive model may serve as a basis for individual risk calculation algorithm prior to delivery. We suggest that in the future, this type of planning of the delivery will serve different health systems, in both high- and low-resource environments, along with the NICU admission or transfer policy.

## 1. Introduction

Neonatal intensive care unit (NICU) admission entails risks for newborns and stress to their families, at a high cost to the healthcare system. The majority of studies that have evaluated admission and risk factors for the NICU use have focused primarily on preterm infants. In light of well-established neonatal bed crises and increasing healthcare costs [1,2], analyses of NICU admissions for term neonates has become imperative. In a time trend analysis of 38 units in the United States between 2007 and 2012, the overall NICU admission rate increased by 23% after adjusting for maternal and neonatal characteristics, while the admitted neonates were increasingly likely to be at term [3]. This trend was further reinforced by the results of other recent studies showing that infants born at term (≥37 weeks’ gestation) account for 53–65% of NICU admissions [3,4,5]. Notably, some have suggested term NICU admission rates as a marker of obstetrical care quality [6].

To our knowledge, no previous studies have comprehensively examined the admission of term infants to NICUs, using an integrated statistical approach. Thus, the primary aim of this study was to identify and quantify risk factors and causes for the NICU admission of term neonates in order to develop a predictive model for use during labor to identify births at high risk for NICU admission, prior to delivery.

## 2. Methods

### 2.1. Study Population and Data Collection

We conducted a retrospective cohort study using the computerized medical record database of a single large obstetric center between 2005 and 2019, with a mean of 13,500 deliveries per year. We included all singleton term live births ≥37 weeks’ gestation. The following were excluded from the study: multifetal birth, preterm birth (<37 weeks’ gestation), and stillbirth (intra-uterine fetal death). Data on demographic and obstetric characteristics as well as data on the course of delivery and delivery complications, maternal blood sample results at admission, and neonatal data were derived from the electronic database management software, which is updated during labor and validated by the computer system’s personnel periodically. The data file was de-identified by the hospital personnel and constructed for analysis.

Coverage for all women and neonates for antenatal, intrapartum, NICU care, and any health costs throughout life is provided under the National Health Plan.

The study protocol was submitted to the institutional Ethical Committee (Helsinki Committee) and was exempted on the basis of an anonymous analysis (reference number SZMC_0199-19); identifying numbers were erased after the data were linked.

### 2.2. Data Collection and Variables

From the electronic medical record, we obtained the birth year, gestational age at delivery, data on NICU use, and maternal characteristics, including age, race and/or ethnicity, insurance, gravidity, parity, mode of delivery, hypertension, and diabetes in pregnancy. Gestational age was based on the best obstetric estimate by using last menstrual period and ultrasound data and was recorded in completed weeks. All births were analyzed, i.e., we included repeated births for the same women. All analyses were performed before and after exclusion of congenital malformations and chromosomal and genetic syndromes.

Demographic and clinical maternal data included maternal age, education, population group, ethnicity, pregnancy complications, and maternal comorbidities (gestational/pre-gestational diabetes, hypertensive disorders, and other background maternal diseases). Current and past obstetrical data included parity, assisted reproductive techniques (ART) used to achieve pregnancy, prior cesarean section, prior miscarriage, high-risk hospitalization during pregnancy, inter-pregnancy interval, antenatal care, past neonatal death, gestational age at birth (recorded by LMP; US exam < 12 weeks gestation or both), admission vital signs (diastolic and systolic blood pressure, temperature (oral), heart rate), CBC and blood type at admission, mode of labor initiation (spontaneous, induction, no trial of labor), mode of delivery (vaginal spontaneous, instrumental or cesarean section), epidural analgesia, oxytocin induction/augmentation, thick meconium, head presentation, normal fetal monitor, modified Bishop score at admission [7], total labor duration (min), stage II duration (min), stage III duration (min), on-call hours at birth, and severe placental, uterine, and maternal complications (dehiscence, uterine rupture, placenta accrete, massive bleeding, post-partum hemorrhage, blood products, re-laparotomy and uterine atony). Neonatal characteristics included birth weight, sex, 1′- and 5′-Apgar score, macrosomia (birth weight [BW] > 4000 g), small for gestational age (SGA), population growth curves [8], and NICU admission (any transfer with a length of stay of at least 4 h), congenital syndromes, and malformations. The diagnoses at admission (one or more) are listed as per ICD coding and diagnoses registered on the neonatal discharge records (three authors: RF, NS, SGG independently reviewed the diagnoses and ICD codes and reached agreement on disease categories; disagreements were discussed jointly and decided): respiratory distress and disorders, meconium aspiration, cardiovascular disorders and hemodynamic instability, 5′-Apgar score < 7 or arterial umbilical blood pH < 7.1, brain morbidities and hypoxic encephalopathy, hypoglycemia, electrolyte imbalance, neonatal jaundice, infection, digestive disorders, musculoskeletal disorders, structural congenital malformations, and chromosomal abnormalities.

In our institution, we routinely admit to the NICU all infants born at <36 weeks of gestation or with birth weight < 2000 g. The decision to admit to the NICU for medical reasons is taken by a senior physician; a staff neonatologist or a neonatology fellow. For the purpose of the study, due to the long duration of the study and possible policy and facility variations, that might determine the NICU admission decision-making process (versus observation at the neonatal maternity ward), we combined the direct delivery room-NICU admission and delayed maternity–NICU admission. The NICU admissions do not include any post-discharge (home) re-admissions; those are reared to the PICU or the pediatric ward. In the present study we decided to limit the inclusion criteria to the standard definition of term since this is the center threshold for a medically indicated NICU admission; otherwise, any newborn at 34–36 weeks gestation is transferred to the NICU for observation, with or without a medical indication for 4 h and without additional clinical judgment.

For the purpose of the study, due to the long duration of the study and possible policy and facility variations that determine the transfer policy, we combined data on direct delivery room–NICU transfer and delayed maternity–NICU transfers (data on separate early/delivery room and delayed NICU transfer were culled accordingly, not reported here).

### 2.3. Data Analysis

#### 2.3.1. Statistical Analyses

Subjects were divided into two groups: (1) Admitted to NICU after delivery; (2) Not admitted to NICU after delivery (no-NICU). Univariate analysis was initially conducted; continuous variables were analyzed using a T-test or Mann–Whitney U-test, and categorical variables were analyzed using Chi-square test or Fisher’s exact test. Multivariable logistic regression models were then carried out in order to identify independent risk factors for NICU admission among term neonates, including demographic, clinical, and obstetrical features. Models were adjusted for delivery and newborn characteristics: congenital malformations and genetic syndromes.

All tests were two-sided; *p*-values < 0.05 were considered statistically significant. Statistical analyses were performed using SPSS version 25.0 statistical package (Armonk, NY, USA, IBM Corp.).

#### 2.3.2. Machine Learning and Gradient Boosting Model

Machine learning (ML) algorithms are built with a focus on automatically analyzing large data sets with large sets of features. The algorithms allow for learning high-order interactions within features without a priori selection of which features are significant. For the current study, the CatBoost gradient boosting model (GBM) was used to predict an individualized risk score for NICU admission among term neonates. This method is based on an ensemble of decision trees built sequentially, with each tree putting additional focus on the mistakes made in the previous decision trees. GBM has been previously used in obstetrical data analyses [9]. Model accuracy was measured via the Area Under the Receiver Operating Characteristic (ROC) Curve (AUC) and validated via 10-fold cross-validation. Confidence intervals were calculated using DeLong’s test [10]. All pre-delivery and during-delivery features were entered into the model. Since this model also uses features with incomplete data, a median imputation for cases of incomplete data was used.

Further ML analyses were performed by SHAP (Shapley additive explanation) in order to identify and interpret the importance of the top influencing features in the GBM and the interactions between specific features [11].

Subjects were then stratified, based on GBM risk score, into low and high risk for term NICU admission. The risk thresholds for high- and low-risk groups were determined using Youden’s Index [12] in order to achieve an optimal balance between sensitivity and specificity. The odds ratios (ORs) for maternal and neonatal adverse outcomes were then calculated according to the GBM risk stratification.

A multivariable regression statistical approach was applied to check for significance, magnitude and direction of the associations identified by the GBM. An accuracy correlation between the logistic regression and GBM was assessed by Spearman’s correlation.

Machine learning was performed using Python v3.6.3 (Beaverton, OR, USA, Python Software Foundation), scikit-learn library v0.20.0 (Beaverton, OR, USA, Python Software Foundation) and CatBoost version 0.15.1 (Moscow, Russia, Yandex LLC).

## 3. Results

### 3.1. Study Population

During the study period 206,509 births were recorded at the Shaare Zedek Medical Center. After application of study exclusion criteria, we included 192,527 (93.23%) term singleton live births in the study (91,697 women): 5292 (2.75%) were admitted to the NICU and 187,235 (97.25%) were not (Figure 1). The NICU admission rate during the study period ranged between 2.1% and 4.1% (Appendix A).

### 3.2. Neonatal Admission to NICU

A comparison between both study groups showed that the NICU admission group differed significantly based on maternal demographic characteristics, clinical and obstetric history, vital signs upon admission for delivery, and delivery and neonatal characteristics (*p* < 0.001 for all) (Table 1). Notably, the mean gestational age at birth was significantly lower, i.e., early term, for the NICU admission group than the no-NICU group (39.1 ± 1.4 weeks vs. 39.5 ± 1.2; *p* < 0.001). The diagnoses at admission (presented as rate of the total term neonates admitted to the NICU) were respiratory distress and disorders 47.1%, structural congenital malformations 44.7%, cardiovascular disorders and hemodynamic instability 20.3%, hypoglycemia 18.7%, 5′-Apgar score < 7 or arterial umbilical blood pH < 7.1 11.7%, neonatal jaundice 11.0%, musculoskeletal disorders 10.9%, infections 7.7%, meconium aspiration 7.1%, brain morbidities and hypoxic encephalopathy 5.5%, chromosomal abnormalities 3.3%, electrolyte imbalance 2.3%, and digestive disorders 0.001%. The blood transfusion rate for the term newborns admitted to the NICU was 1.2% and 0.028% for the entire study population. We further dissected the term period in order to quantify the risk variation for each week of gestation and observed a peak rate of NICU admission during the 37th week of gestation (8.4%), which decreased significantly thereafter, followed by an additional rise at 42 (3.1%) (Figure 2A). For neonates born between 37 + 0 and 37 + 6 weeks’ gestation we found that the highest rate of admission was for those born at 37 weeks + 0 days (12.7%), with a steady decline to the lowest admission risk of 5.2% at 37 weeks + 6 days (*p* for trend < 0.001) (Figure 2B).

An adjusted multivariable logistic regression model was used to assess the risk factors for NICU admission for term neonates. In this model, the significant risk factors were nulliparity (OR = 1.19, 95%CI [1.07, 1.33]; *p* = 0.002); pregnancy complications: hypertensive disease and gestational diabetes mellitus (OR, 95%CI 1.28 [1.02, 1.60]; *p* = 0.035 and 2.52 [2.09, 3.03]; *p* < 0.001 respectively); hospitalization during pregnancy (OR = 1.42, 95%CI [1.21, 1.66]; *p* < 0.001); induction of labor (OR = 1.28, 95%CI [1.12, 1.46]; *p* < 0.001); elective cesarean section (OR = 1.52, 95%CI [1.05, 2.22]; *p* < 0.001); and gestational age at birth of 37 weeks (OR = 2.99, 95%CI [2.63, 3.41]; *p* < 0.001) with an AUC of 0.723 (95%CI [0.712–0.735]; *p* < 0.001), Table 2. Importantly, the pattern remained significant after adjustment for congenital malformations and genetic syndromes.

A GBM for predicting NICU admission was set in a 10-fold cross-validation study. The model included all features collected pre-delivery and during delivery, prior to birth, and achieved an AUC of 0.750 (95%CI 0.743–0.757). The ranking of the top ten significant features, in order of importance, was: spontaneous delivery, labor duration, thick meconium, gestational age by day, normal labor progress, first maternal white blood cell count (admission), gestational age by week, abnormal labor progress, cesarean section delivery, and admission week, Figure 3.

The NICU admission prediction risk assessment was compared between the logistic regression and GBM models (Spearman correlation calculation), achieving an R = 0.667 (*p* < 0.001).

Thus, we used the GBM to develop a standard for a clinical decision algorithm and future planning for term neonate NICU admission; the study population was divided into high and low risk for NICU admission using the GBM individualized prediction risk score. Youden’s Index threshold (4%) was applied, and the population was stratified with 27% (n = 50,537) allocated to the high-risk group and 73% (n = 141,990) to the low-risk group. Among those classified by our model as high risk, 64% were admitted to the NICU after delivery. The risk for NICU admission among the high-risk group was 5.26-fold higher as compared to those designated as low risk by our model. In order to further substantiate our stratification, the odds ratios for adverse outcomes (maternal and neonatal) were calculated by comparing the outcome incidence in the designed high-risk group using the low-risk group as reference, Table 3. The neonates designated as being at high risk of NICU admission had increased risks for maternal and neonatal adverse outcomes. Severe maternal uterine and placental pathologies were found to have the heaviest weight odds ratios for the neonatal NICU admission high-risk group as designated by the GBM, including uterine dehiscence (OR = 43.52), uterine rupture (OR = 23.13), placenta accreta (OR = 14.05), and massive antenatal/ intrapartum bleeding (OR = 8.71).

## 4. Discussion

Previous studies have focused on specific maternal and neonatal characteristics associated with NICU admission for term neonates, including maternal diseases and therapies [6], mode of delivery [13,14], birth weight [15,16], delivery complications and neonatal diseases [17,18,19,20], breastfeeding, and mother–child bonding [21,22]. In this study, we examined risk factors for NICU admission in a large single-center cohort of term neonates. We show that the use of ML models allowed us to rank the risk factors and identify individual risk based on a pre-birth prediction model for NICU admission, which correlated well with the maternal and neonatal adverse outcome prediction.

The study population included 192,527 term neonates, with a 2.75% rate of NICU admission following delivery. Compared to other large-scale studies conducted [6,14,17,18,19,20], this admission rate for NICU lays within the lower range; in another study, the risk for NICU admission was 4.1 percent, and risk factors for admission included non-citizen status, low or no health insurance coverage, and premature rupture of membranes [23]. We suggest that the difference in NICU admission rate may be explained mainly by our study being performed at a single center, with a homogenous population covered by a national healthcare plan, no out of hospital transfers, and a highly trained neonatology service present in the delivery room. In addition, different exclusion criteria and different hospitalization protocols (e.g., induction of preterm premature rupture of membranes prior to term, at 35 weeks’ gestation) could partially explain the difference. We noted a recent increase in the NICU admission rate in the later period of the study that paralleled a large-scale building initiative and center development. This resulted in an increased availability of NICU beds and perhaps made the delivery room/maternity transfer decisions easier, probably leading to more infants with mild respiratory distress in the transition time (the immediate period after delivery) being transferred to the NICU. The main diagnoses by others for a term infant at admission to the NICU are: hypoglycemia, respiratory distress, cardiovascular instability, and hyperbilirubinemia [24]. Our population showed a similar pattern for diagnoses at the time of the NICU admission. Notably, the transfusion of blood products in critically ill newborns is important and has been a hot topic in recent years. The need for such a transfusion may reflect hemodynamic instability or an ominous in-utero condition, and as such, affects the immediate and long-term prognosis for the neonate. Since our manuscript focused on predicting NICU admission and risk factors at admission for term neonates, we did not include data regarding transfusion, as this would be a manifestation of the underlying disease and would not be the cause of admission per se. In our data, the overall blood transfusion rate for term neonates admitted to the NICU was extremely low, overall, as well as those admitted to the NICU. We assume that this is a rare event for term neonates for which the main indication for blood products transfusion is severe neonatal jaundice/exchange and rare causes of anemia. However, due to the large population in our study, we were able to comprehensively assess maternal background, vital signs and blood tests at admission for birth, characteristics of labor, and neonatal characteristics, which were included in the ML NICU admission risk stratification model. Nevertheless, our study identified nulliparity, pregnancy complications (hypertensive disease and gestational diabetes mellitus), hospitalization during pregnancy, mode of labor initiation, and gestational age at birth of 37 weeks as pre-delivery significant factors for predicting risk of NICU admission.

The strong association between gestational age at birth and the risk for NICU admission questions the definition of “term pregnancy” (≥37 weeks). The present study focused on medically indicated NICU admission for term neonates. Since different institutions use different thresholds for routine admission of near-term infants, we only included term neonates born after 37 + 0 weeks of gestation. Notably, in this population, the risk for NICU admission peaked at 37 + 0 weeks and declined with each additional day within the 37th week, continuously decreasing, reaching a nadir at 39 to 40 weeks’ gestation, followed by an increase reaching a second peak at 42–43 weeks’ gestation. The antenatal care, including US assessment, for the study population is covered by the National Health Insurance plan; the gestational age based on LMP and US is <12 weeks. This minimizes the error in the calculation of the gestational age at birth. Traditionally, infants born after 37 completed weeks’ gestation are classified as “term”. It has been shown that health outcomes vary widely according to week of gestation; morbidities such as respiratory distress syndrome (RDS), transient tachypnea of the newborn (TTN), and hypoglycemia are more common among early term infants, and incidence decreases with increasing gestational age [13,19]. Furthermore, infants born at 34–36 weeks of gestation, considered late pre-term, are more likely to require admission to NICUs [25]. The time elapsed within the 37th week of gestation remains vaguely defined. The American College of Obstetrics and Gynecology (ACOG) has suggested subdividing infants into groups of early term (37–39 weeks), term (39–41 weeks), late term (41–42 weeks), and post-term (42+ weeks) [26,27]. Our study strongly supports this classification and indicates that neonates born at 37 weeks’ gestation have a risk of NICU admission that is similar to those born at 36 weeks’ gestation rather than term (>38 weeks’ gestation). This is in accordance with other studies that showed that the elimination of early term delivery programs did not lead to a reduction in NICU admissions but were associated with fewer short NICU stays [13]. We consider that this association with NICU admission risk represents an additional milestone for re-definition of “term” neonates. However, intrapartum sentinel events such as uterine rupture, heavy meconium, and placental abruption should not be underscored as main determinants for NICU admission [28,29].

By using the GBM, our goal was to identify the best predictors for term NICU admission without initial processing, as used in regression models, and at different time points, including before labor and after labor outcome. We identified various pre-labor and during-labor features which were correlated with NICU admission; the ML prediction model that included all data reached a high positive predictive value for each of the features with a high predictive value for the high-risk group. Others have suggested predictive models relating to the NICU population using ML; however, these were not limited to the term neonatal population and mainly focused on NICU admission trends among different epochs and neonatal outcomes during NICU admission [30,31,32]. We based our tandem statistical approach based on previous studies which demonstrated that the machine learning methodology was able to determine additional significant medically relevant information for the prognosis, diagnosis, and therapy of several morbidities [33,34]. In our study, we demonstrated that the traditional statistical method risk assessment was highly correlated [R = 0.667 (*p* < 0.001)] to the machine learning approach. However, we also demonstrated the superiority of the machine learning approach with an AUC of 0.750 vs. AUC of 0.723, the predictive value of the traditional model. Furthermore, we described another benefit of using the machine learning approach: specifically in the calculation of the relative weight and ranking of each risk factor. We believe this tandem approach of traditional statistics and machine learning provides a template that could be used by others in future research endeavors. Specifically, the application of machine learning to critical care data may provide important assistance in the understanding, prediction, and ranking of critical illness risk factors in infants.

Although the risk assessment correlation between the traditional statistical methods and ML was high, we postulate that using machine learning for the determination of risk factors alone is not only additive to the traditional statistical methods. ML may be used for the calculation of the relative weight and ranking of each risk factor, thus allowing more appropriate risk algorithm production. The issue of risk factors identification and ranking by ML alone in the medical field still requires additional studies in a large and variable population. The present study that used the tandem approach of the multivariate analyses and ML models in the analysis of risk assessment and risk group definition is an example for the future use of this tandem analyses while exploiting the advantages of each approach [35,36].

## 5. Strengths and Limitations

This study adds to a limited number of studies which examined risk factors and predictive models for term neonates’ NICU admission. The present study focuses on the largest population of neonates, term neonates. It was designed as a large and high-quality study based on validated linked databases within a single institution. This enabled us to adjust our analyses for congenital malformations and genetic syndromes. The use of a single center and neonates born in-hospital offers the advantage of uniform management care, both for obstetrics and neonatology. The antenatal, labor, and all health and hospitalization costs are covered by the National Health Insurance Plan for all mothers and neonates. All of these properties allow us to assess the reliability of the results and reach robust conclusions. Additionally, although the tandem use of multivariate analyses and ML in medicine is emerging, it is currently not widely used for the prediction of NICU admission among neonates and risk ranking and classification; therefore, we consider this study to be an initiator.

Nevertheless, some limitations need to be considered. Firstly, Shaare Zedek Medical Center is located in Jerusalem and serves a relatively homogeneous population, consisting mainly of religious and ultra-Orthodox Jews; therefore, it is not necessarily possible to project our results onto other populations, and therefore, it may have low external validity. However, the great variety of NICU cases and diagnoses is similar to other reports, thus partially mitigating this limitation. In order to better discuss the term definition for newborns, we lack the data on neonates less than 36 weeks; however, this was beyond the aims of the study. In addition, we lacked data on maternal nutrition supplements during pregnancy, BMI, and smoking status, which may have affected NICU admission. However, our population care is covered by the National Health Insurance plan, which is characterized by more than 80% attendance for early pregnancy care and free access to prenatal nutritional supplements.

## 6. Conclusions

This study integrates multivariate analyses with ML models in the NICU admission risk identification and ranking for term neonates, at different points in time, and especially before birth. Other medical facilities may apply our ML risk ranking and thresholds for the assessment of each individual neonate before birth, thereby tailoring the need for intensive care.

## Figures and Tables

**Figure 1 jcm-11-04258-f001:**
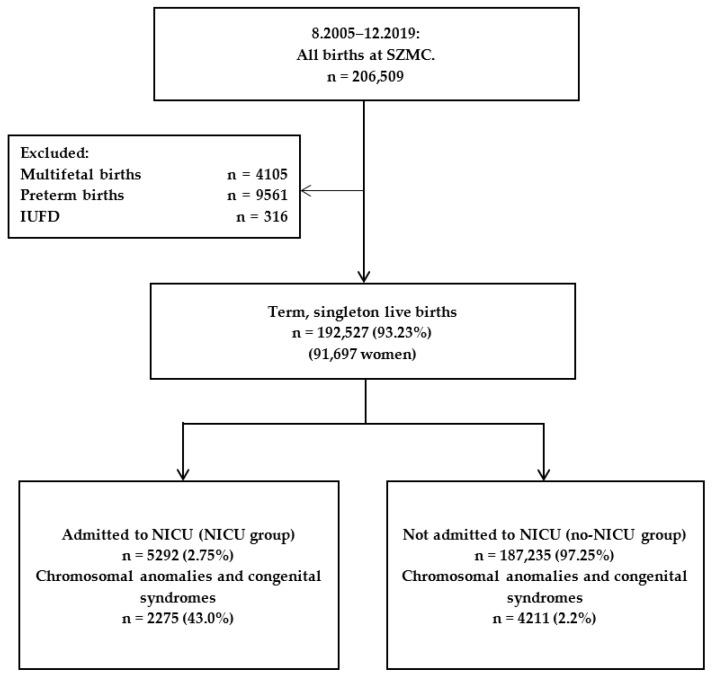
Study population flow diagram.

**Figure 2 jcm-11-04258-f002:**
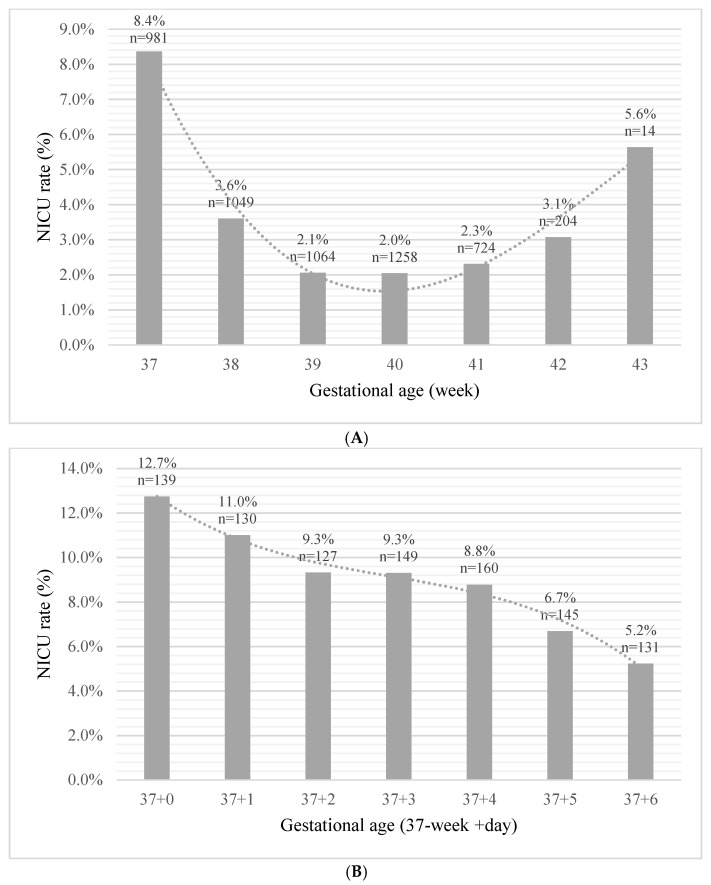
(**A**): NICU admission rate by week of gestation at birth. (**B**): NICU admission rate within the 37th week.

**Figure 3 jcm-11-04258-f003:**
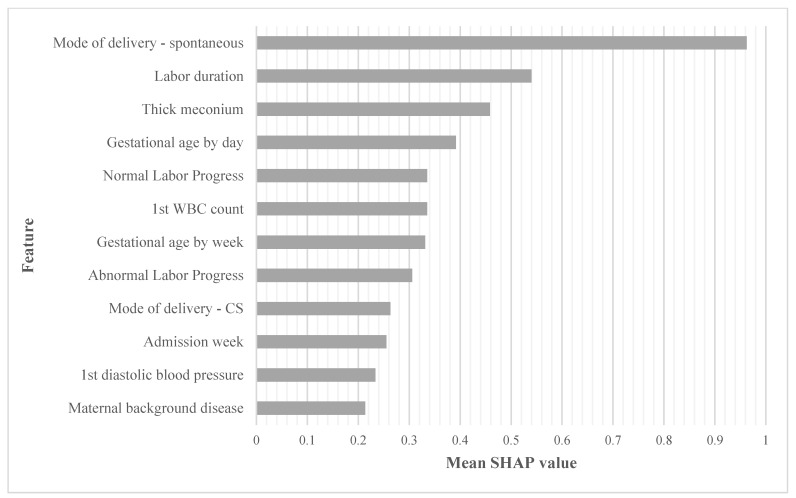
Risk ranking using a gradient boosting model to identify the most important influencing features in order of relative importance. The top most influential features according to the GBM, by order of importance. As seen, spontaneous delivery was the most influential feature. Values are measured by relative importance based on shapely additive explanation (SHAP). WBC, white blood cells; CS, cesarean section; SHAP, shapely additive explanation.

**Table 1 jcm-11-04258-t001:** Demographic, obstetrical, and neonatal characteristics of the study groups.

	Characteristics	NICU (n = 5292)	No-NICU (n = 187,235)	*p*
Demographics and clinical history	Maternal age, mean (SD), Y	29.3 (6.2)	28.9 (5.7)	<0.001
	Maternal age > 35, n (%)	988 (18.7)	28,139 (15.0)	<0.001
	Arab population group, n (%)	590 (11.1)	14,514 (7.8)	<0.001
	Ethiopian ethnicity, n (%)	101 (1.9)	2045 (1.1)	<0.001
	Education > 12 years, n (%)	4884 (96.3)	175,329 (97.8)	<0.001
	GDM/DM, n (%)	544 (10.3)	6127 (3.3)	<0.001
	Hypertensive disorder, n (%)	246 (4.6)	3500 (1.9)	<0.001
	Maternal background disease, n (%)	1441 (27.2)	28,253 (15.1)	<0.001
Obstetrical history	Gestational age at birth, mean (SD), W	39.1 (1.4)	39.5 (1.2)	<0.001
	Parity, mean (SD), N	3.3 (2.6)	3.5 (2.4)	<0.001
	ART, n (%)	249 (4.7)	5390 (2.9)	<0.001
	Prior CS, n (%)	1047 (19.8)	21,439 (11.5)	<0.001
	Prior miscarriage, n (%)	1581 (29.9)	54,562 (29.1)	0.25
	High risk hospitalization in pregnancy, n (%)	327 (8.1)	6977 (4.7)	<0.001
	Inter-pregnancy interval, mean (SD), W	94.7 (73.9)	85.1 (61.2)	<0.001
	Past neonatal death, n (%)	46 (0.9)	877 (0.5)	<0.001
	Antenatal care, n (%)	4282 (80.9)	154,673 (82.6)	<0.001
Obstetrical data	Mode of delivery, n (%)			<0.001
	Spontaneous	2959 (55.9)	159,730 (85.3)	
	Instrumental	674 (12.7)	10,735 (5.7)	
	CS	1659 (31.3)	16,770 (9.0)	
	Mode of labor initiation, n (%)			<0.001
	Spontaneous	3363 (63.5)	159,696 (85.3)	
	Induction	997 (18.8)	17,165 (9.2)	
	Elective CS	932 (17.6)	10,374 (5.5)	
	TOLAC, n (%)	479 (9.1)	14,661 (7.8)	0.001
	Thick meconium, n (%)	578 (10.9)	9030 (4.8)	<0.001
	Head presentation, n (%)	5034 (95.4)	183,112 (98.0)	<0.001
	Labor duration, mean (SD), Min	442.1 (563.0)	271.5 (344.5)	<0.001
	Stage II duration, mean (SD), Min	50.5 (63.6)	33.0 (49.2)	<0.001
	Stage III duration, mean (SD), Min	11.6 (9.2)	10.7 (26.3)	<0.001
	On-call hours at births, n (%)	3532 (66.7)	134,621 (71.9)	<0.001
Neonatal characteristics and outcomes	Female sex, n (%)	2273 (43.0)	91,595 (48.9)	<0.001
	Birth weight, mean (SD), Gr	3222.6 (576.4)	3333.6 (424.5)	<0.001
	5’-Apgar ≤ 7, n (%)	794 (15.0)	2015 (1.1)	<0.001
	Macrosomia, n (%)	422 (8.2)	11,751 (6.3)	<0.001
	SGA, n (%)	844 (15.9)	11,224 (6.0)	<0.001
	Pediatrician present at birth, n (%)	1492 (28.2)	14,154 (7.6)	<0.001

Data presented as the group means (SD) or as frequencies (percentages). Abbreviations: GDM/DM, gestational/pregestational diabetes mellitus; ART, Assisted reproductive technology; CS, cesarean section; TOLAC, trial of labor after cesarean; SGA, small for gestational age. Y, years; W, weeks; N, number; Min, minutes; Gr, grams.

**Table 2 jcm-11-04258-t002:** Neonatal intensive care admission predictors, adjusted for gestational age at birth and neonatal characteristics.

	OR [95%CI]	*p*
Maternal age >35 years	1.06 [0.94, 1.21]	0.32
Arab population group	1.13 [0.98, 1.31]	0.081
Ethiopian ethnicity	1.01 [0.69, 1.46]	0.96
GDM/DM	2.52 [2.09, 3.03]	<0.001 *
Hypertensive disorder	1.28 [1.02, 1.60]	0.035 *
ART	0.88 [0.70, 1.11]	0.29
Maternal background disease	1.15 [1.02, 1.31]	0.022 *
Antenatal care	0.89 [0.80, 1.01]	0.079
Nulliparity	1.19 [1.07, 1.33]	0.002 *
Prior CS	1.08 [0.93, 1.26]	0.32
High risk hospitalization	1.42 [1.21, 1.66]	<0.001 *
Head presentation	0.91 [0.57, 1.44]	0.69
On-call hours at birth	0.95 [0.86, 1.04]	0.29
Gestational age at birth = 37 W	2.99 [2.63, 3.41]	<0.001 *
Gestational age at birth 42–43 W	0.98 [0.80, 1.22]	0.91
Induction	1.28 [1.12, 1.46]	<0.001 *
Elective CS	1.52 [1.05, 2.22]	0.026
Education > 12 years	1.04 [0.77, 1.39]	0.81

Significant differences are indicated by an asterisk (*). Model was adjusted to gestational age at birth and neonatal characteristics. Abbreviations: OR, odds ratio; CI, confidence interval; GDM/DM, gestational/pregestational diabetes mellitus; ART, Assisted reproductive technology; CS, cesarean section; W, weeks.

**Table 3 jcm-11-04258-t003:** Maternal and neonatal adverse outcomes: high- and low-risk groups for NICU admission.

	Feature	OR [95%CI]	*p*-Value	Crude Numeric Results *
Severe placental, uterine, and maternal bleeding complications	Uterine Dehiscence	43.52 [26.65, 71.09]	<0.001	[141,973, 50,275], [17, 262]
	Uterine rupture	23.13 [13.28, 40.29]	<0.001	[141,976, 50,422], [14, 115]
	Placenta Accreta	14.05 [5.38, 36.71]	<0.001	[141,985, 50,512], [5, 25]
	Massive antenatal/intrapartum bleeding	8.71 [5.32, 14.24]	<0.001	[141,969, 50,472], [21, 65]
	PPH	2.32 [2.24, 2.41]	<0.001	[135,199, 45,254], [6791, 5283]
	Blood products transfusion	3.74 [3.41, 4.11]	<0.001	[141,203, 49,504], [787, 1033]
	Re Laparotomy	2.14 [1.79, 2.55]	<0.001	[141,703, 50,319], [287, 218]
	Uterine atony	1.56 [1.21, 2.03]	0.007	[141,830, 50,448], [160, 89]
	NICU admission	5.26 [4.96, 5.56]	<0.001	[140,079, 47,156], [1911, 3381]
Severe neonatal complications	Meconium aspiration	7.59 [5.71, 10.09]	<0.001	[141,925, 50,362], [65, 175]
	Infection	5.56 [4.20, 7.35]	<0.001	[141,916, 50,391], [74, 146]
	Hypoxic ischemic encephalopathy (high grade)	3.36 [2.76, 4.07]	<0.001	[141,801, 50,312], [189, 225]
	Chromosomal anomaly	3.14 [2.48, 3.99]	<0.001	[141,862, 50,394], [128, 143]
	GIT complications	2.41 [1.54, 3.76]	0.001	[141,948, 50,501], [42, 36]
	Electrolytes disorders	2.81 [1.58, 5.01]	0.005	[141,967, 50,514], [23, 23]
	Congenital malformations	1.28 [1.14, 1.45]	0.001	[141,142, 50,151], [848, 386]
Apgar scores	1′-Apgar1 ≤ 7	3.95 [3.75, 4.16]	<0.001	[139,396, 47,076], [2594, 3461]
	5′-Apgar ≤ 7	1.84 [1.70, 1.99]	<0.001	[140,286, 49,432], [1704, 1105]
Congenital disorders	Chromosomal (numeric)	3.26 [2.52, 4.21]	<0.001	[141,882, 50,412], [108, 125]
	Genetic (non-numeric chromosomal and others)	2.91 [1.75, 4.84]	<0.001	[141,961, 50,507], [29, 30]
	SGA	1.58 [1.52, 1.65]	<0.001	[134,186, 46,273], [7804, 4264]
Pediatric re-admission	≤7 days	2.31 [1.91, 2.78]	<0.001	[141,747, 50,338], [243, 199]
	≤14 days	2.43 [2.09, 2.82]	<0.001	[141,615, 50,214], [375, 323]
	≤21 days	2.47 [2.14, 2.85]	<0.001	[141,580, 50,178], [410, 359]
	≤42 days	2.34 [2.06, 2.67]	<0.001	[141,494, 50,125], [496, 412]
	≤90 days	2.11 [1.87, 2.37]	<0.001	[141,348, 50,058], [642, 479]

The reference category is low-risk for NICU admission group. * Crude numeric results [low risk without feature, high risk without feature], [low risk with feature, high risk with feature]. Abbreviations: OR, odds ratio; CI, confidence interval; NICU, neonatal intensive care unit; PPH, post-partum hemorrhage; GIT, gastro-intestinal tract; SGA, small for gestational age.

## Data Availability

The data was retrieved from the EMR of Shaare Zedek Medical Center; available upon official request from Shaare Zedek Medical Center R&D Authority.

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
