# Peer review of "NICU Admission for Term Neonates in a Large Single-Center Population: A Comprehensive Assessment of Risk Factors Using a Tandem Analysis Approach"

_jcm, 2022, doi:10.3390/jcm11154258_

Round 1

Reviewer 1 Report

Thank you for submitting the manuscript. I read your paper with great interest. The issue is certainly important as the management of intensive care, especially in the case of limited resources, is an important issue in every hospital.I would like you to provide more considerations on the aspect of the need for transfusions in the newborn. In fact, it is a debated topic that has repercussions in terms of prognosis.I suggest some references:

doi: 10.1097/EJA.0000000000001646.

doi: 10.1111/pan.14457.

doi: 10.1002/14651858.CD004863.pub2.

I hope my comments are useful to you.

Kind Regards

Author Response

Thank you for your comment. Indeed, transfusion of blood products in critically ill newborns is important and has been a hot topic in recent years. The need for such a transfusion may reflect hemodynamic instability or an ominous in- utero condition; and as such effect the immediate and long-term prognosis for the neonate. Since our manuscript focused on predicting NICU admission and risk factors at admission for term neonates, we did not include data regarding transfusion; as this would be a manifestation of the underlying disease and would not be the cause of admission per se. In our population, the overall blood transfusion rate for term neonates admitted to the NICU was 1.2%. Since for all term neonates in our center the rate of blood products transfusion is extremely rare, 0.028%; we postulate that this is a rare event for term neonates in general, as well as for those admitted to NICU for which the main indication for blood products transfusion is severe neonatal jaundice/exchange and rare causes of anemia.

This is stated now in the Results section, lines 190-192 and Discussion section, lines 290-301.

Reviewer 2 Report

The authors conducted a retrospective analysis using a large single center data to identify risk factors for NICU admission using a Tandem analysis approach. The topic is important since stratifying risk factors for NICU admission could allow clinician to intervene mothers or babies if the factors are modifiable. However, I have some comments for publication.

Comments

1.

I agree with the idea that machine learning may be a useful new statistical method to produce risk algorithm. However, “the aim of the study was to identify and quantify risk factors and causes of NICU admission of term neonates” not to evaluate a new statistical method. I suppose readers would want to know why a new statistical approach is needed to achieve the aim? Unless a new method is proven to be more excellent than another conventional method, the interpretation of the result can be difficult for readers, since I guess most of the readers are clinicians who do not have enough knowledge on this new statistical method.

2.

I agree that the risk for NICU admission peaked at 37+0 weeks based on the data of patients between 37 to 42 weeks. However, the data on neonate less than 36 weeks is informative if the authors want to discuss on the threshold of gestation on term neonate.

3.

Is there any local NICU admission criteria in the center in which the research conducted? This could modify the result.

4.

Data on actual reason for NICU admission (such as respiratory distress, hemodynamic instability etc.) would be informative from clinician’s perspective. This is because these data allow clinician to predict what kinds of intervention can be available to improve outcomes.

5.

Is year of born associated with NICU admission rate? This is because as time goes by the management of neonatal care is improving, and it may affect the outcomes.

Author Response

  1. I agree with the idea that machine learning may be a useful new statistical method to produce risk algorithm. However, “the aim of the study was to identify and quantify risk factors and causes of NICU admission of term neonates” not to evaluate a new statistical method. I suppose readers would want to know why a new statistical approach is needed to achieve the aim? Unless a new method is proven to be more excellent than another conventional method, the interpretation of the result can be difficult for readers, since I guess most of the readers are clinicians who do not have enough knowledge on this new statistical method.

Answer:

Indeed, this is a relative new approach; the method is not canonical and not intuitive for the clinician. However, in time and with additional studies that will use it, similar with other traditional statistics methods, we believe that clinicians will find it helpful, especially in the ranking the risk factors (for example Fig 3). We now elaborate in the manuscript:

We based our tandem statistical approach based on previous studies which demonstrated that the machine learning methodology was able to determine additional significant medically relevant information for prognosis, diagnosis, and therapy of several morbidities (Ref number 33, 34). In our study we demonstrated that the traditional statistical method risk assessment was highly correlated [R=0.667 (P<0.001)] to the machine learning approach. However, we also demonstrated the superiority of the machine learning approach with an AUC of 0.750 vs. AUC of 0.723; the predictive value of the traditional model. Furthermore, we described another benefit of using the machine learning approach; specifically in the calculation of the relative weight and ranking of each risk factor. We believe this tandem approach of traditional statistics and machine learning provides a template that could be used by others in future research endeavors. Specifically, application of machine learning to critical care data may provide important assistance in the understanding, predicting and ranking of critical illness risk factors in infants (Discussion section, lines 346-359).

  1. I agree that the risk for NICU admission peaked at 37+0 weeks based on the data of patients between 37 to 42 weeks. However, the data on neonate less than 36 weeks is informative if the authors want to discuss on the threshold of gestation on term neonate.

Answer:

The present study focused on medically indicated NICU admission for term neonates. Since different institutions use different thresholds for routine admission of near-term infants, we only included term neonates born after 37+0 weeks of gestation in order to add to the external validity of our study. The limitation for the “term definition” is now noted.

This is now stated in the Methods section, lines 120-124 and Discussion/ Limitations section, lines 389-391.

  1. Is there any local NICU admission criteria in the center in which the research conducted? This could modify the result.

Answer:

We now have added the criteria in the Methods section, lines 108-116 (partially overlapping answer to point 2): In our institution we routinely admit to the NICU all infants born at < 36 weeks of gestation or with birth weight < 2000 grams. The decision to admit at NICU for medical reasons is taken by a senior physician; a staff neonatologist or a neonatology fellow. For the purpose of the study, due to the long duration of the study and possible policy and facility variations, that might determine the NICU admission decision making process (versus observation at the neonatal maternity ward), we combined the direct delivery room-NICU admission and delayed maternity-NICU admission. The NICU admissions do not include any post-discharge (home) re-admissions; those are reared to the PICU or the pediatric ward. 

  1. Data on actual reason for NICU admission (such as respiratory distress, hemodynamic instability etc.) would be informative from clinician’s perspective. This is because these data allow clinician to predict what kinds of intervention can be available to improve outcomes.

Answer:

Data concerning the diagnoses at admission are now added in the Methods, Results and Discussion sections as follows:  

Methods section, lines 99-107: The diagnoses at admission (one or more) are listed as per ICD coding and diagnoses registered on the neonatal discharge records (three authors: RF, NS, SGG independently reviewed the diagnoses and ICD codes and reached agreement on disease categories; disagreements were discussed jointly and decided): respiratory distress and disorders, meconium aspiration, cardiovascular disorders and hemodynamic instability, 5’Apgar score < 7 or arterial umbilical blood pH < 7.1, brain morbidities and hypoxic encephalopathy, hypoglycemia, electrolyte imbalance, neonatal jaundice, infection, digestive disorders, musculoskeletal disorders, structural congenital malformations and chromosomal abnormalities.

Results section, lines 184-192: The diagnoses at admission (presented as rate of the total term neonates admitted to the NICU, in order of magnitude) were respiratory distress and disorders 47.1%, structural congenital malformations 44.7%, cardiovascular disorders and hemodynamic instability 20.3%, hypoglycemia 18.7%, 5’Apgar score < 7 or arterial umbilical blood pH < 7.1, 11.7%, neonatal jaundice 11.0%, musculoskeletal disorders 10.9%, infections 7.7%, meconium aspiration 7.1%, brain morbidities and hypoxic encephalopathy 5.5%, chromosomal abnormalities 3.3%, electrolyte imbalance 2.3% and digestive disorders 0.001%.

Discussion section, lines 288-291: The main diagnoses by others, for a term infant at admission to the NICU are: hypoglycemia, respiratory distress, cardiovascular instability and hyperbilirubinemia (Ref number 24). Our population showed a similar pattern for diagnoses at the time of the NICU admission.

  1. Is year of born associated with NICU admission rate? This is because as time goes by the management of neonatal care is improving, and it may affect the outcomes.

Answer:

We analyzed the yearly NICU admission rate for term newborns for the period of the study. This is now provided as Appendix 1 (Histogram). Please see the attachment.

Also, we refer in the results and discussion sections as follows:

Results section, lines 174-175: The NICU admission rate during the study period ranged between 2.1% to 4.1% (Appendix 1). 

Discussion section, lines 283-288: We noted a recent increase in the NICU admission rate in the later period of the study that paralleled a large-scale building initiative and center development. This resulted in an increased availability of NICU beds and perhaps made easier the delivery room /maternity transfer decisions; probably leading to more infants with mild respiratory distress in the transition time (the immediate period after delivery) being transferred to the NICU.

Round 2

Reviewer 2 Report

Thank you for revising the manuscript. The authors responded all of my comment successfully. The manuscript has improved, and I believe the research deserves publication.

Comments

1.Thank you for adding new comment on discussion section regarding the merit of machine leaning. This would be of help for clinicians to understand the merit of this new statistical approach.

2. I understand that all newborn younger than 36w gestation admit to the NICU in the authors’ institution. The comment added on Methods section and Discussion/ Limitations section make this clear.

3/4. Thank you for adding local NICU admission criteria in the Methods section and data concerning the diagnoses at admission. I believe these data is informative for clinicians in terms of external validity.

5. Thank you for analyzing the yearly NICU admission rate for term newborns for the period of the study and adding data on admission rate during the study period.